# Molecular Subtyping and Survival Analysis of Osteosarcoma Reveals Prognostic Biomarkers and Key Canonical Pathways

**DOI:** 10.3390/cancers15072134

**Published:** 2023-04-04

**Authors:** Siddesh Southekal, Sushil Kumar Shakyawar, Prachi Bajpai, Amr Elkholy, Upender Manne, Nitish Kumar Mishra, Chittibabu Guda

**Affiliations:** 1Department of Genetics, Cell Biology and Anatomy, University of Nebraska Medical Center, Omaha, NE 68198, USA; 2Department of Pathology, University of Alabama at Birmingham, Birmingham, AL 35233, USA; 3Center for Biomedical Informatics Research and Innovation, University of Nebraska Medical Center, Omaha, NE 68198, USA

**Keywords:** multiomics data analysis, Cluster-Of-Clusters Analysis, COCA, TARGET database, DNA methylation, classification of osteosarcoma, mRNA and miRNA expression

## Abstract

**Simple Summary:**

Osteosarcoma (OS), which accounts for about 35% of bone malignancy, shows aggressive progression in adults, adolescents, and children. Neoadjuvant chemotherapy provides an improvement in survival for OS patients; however, a molecular level understanding of the disease mechanisms is needed. We utilized publicly available multiomics data from the TARGET database, to perform survival analyses, differential expression analyses, mutational analyses, and subtyping using integrative clustering. Our results have identified several prognostic biomarkers (such as *RAMP1*, *CRIP1*, *CORT*, *CHST13*, and *DDX60L*) in OS patients that can be further explored for therapeutic applications.

**Abstract:**

Osteosarcoma (OS) is a common bone malignancy in children and adolescents. Although histological subtyping followed by improved OS treatment regimens have helped achieve favorable outcomes, a lack of understanding of the molecular subtypes remains a challenge to characterize its genetic heterogeneity and subsequently to identify diagnostic and prognostic biomarkers for developing effective treatments. In the present study, global analysis of DNA methylation, and mRNA and miRNA gene expression in OS patient samples were correlated with their clinical characteristics. The mucin family of genes, *MUC6*, *MUC12*, and *MUC4*, were found to be highly mutated in the OS patients. Results revealed the enrichment of molecular pathways including Wnt signaling, Calcium signaling, and PI3K-Akt signaling in the OS tumors. Survival analyses showed that the expression levels of several genes such as *RAMP1*, *CRIP1*, *CORT*, *CHST13*, and *DDX60L*, miRNAs and lncRNAs were associated with survival of OS patients. Molecular subtyping using Cluster-Of-Clusters Analysis (COCA) for mRNA, lncRNA, and miRNA expression; DNA methylation; and mutation data from the TARGET dataset revealed two distinct molecular subtypes, each with a distinctive gene expression profile. Between the two subtypes, three upregulated genes, *POP4*, *HEY1*, *CERKL*, and seven downregulated genes, *CEACAM1*, *ABLIM1*, *LTBP2*, *ISLR*, *LRRC32*, *PTPRF*, and *GPX3*, associated with OS metastasis were found to be differentially regulated. Thus, the molecular subtyping results provide a strong basis for classification of OS patients that could be used to develop better prognostic treatment strategies.

## 1. Introduction

Osteosarcoma (OS), a highly aggressive bone marrow malignancy that originates in mesenchymal tissue, accounts for about 35% of bone cancer cases worldwide. Other bone tumors include chondrosarcomas (25%), and Ewing sarcomas (16%) [1,2,3,4]. The most affected bones mainly include the humerus (10%), femur (43%), and tibia (23%), while sacrum areas, spine, and pelvis are rarely affected [5,6]. As per National Cancer Institute (NCI) data, for adults, OS (28%) is the second most prevalent primary bone cancer after chondrosarcoma (40%), while for children and adolescents, OS (56%) is the most common followed by Ewing sarcoma (34%). OS predominantly affects males [7,8]. Clinical manifestation suggests that, in 10–30% of the affected patients, metastases occur primarily in lungs (85%) and bones (8–10%) [5,9].

For various cancer types, including OS, genetic heterogeneity due to genomic instability poses challenges to the efficacy of current treatment strategies [10,11,12], demonstrating the need for accurate molecular subtyping to guide targeted therapeutic regimens for OS. Current advances in omics technologies, especially genomics, epigenomics, and transcriptomics, in combination with advanced data analytical tools have enabled molecular classification or subtyping of heterogeneous cancers that are increasingly preferred by physicians over the traditional and error-prone approaches based on clinical characteristics such as the histopathology, grade, and other visual observations [13]. To overcome the limitations of the previous approaches, recent methodologies have focused on the use of molecular profiling data for subtyping of cancers, including those of lung [14,15,16], colon [17,18], breast [19,20,21], and others [22,23,24]. In addition, new techniques such as NanoString and tissue microarray (TMA) approaches have also been evolved for subtype characterization of cancers [25,26,27,28].

In parallel, a variety of statistical and machine learning approaches such as Support Vector Machines (SVMs) and Deep Neural Networks (DNN) have been developed for molecular characterization and subclassification of cancers by assessing multiomics data [25,26]. These studies have been mainly fueled by the ease of access to public data repositories such as International Cancer Genome Consortium (ICGC, https://dcc.icgc.org/, accessed on 1 December 2021), The Cancer Genome Atlas (TCGA), and Gene Expression Omnibus (GEO) for cancer sub-classification studies on the genomic, transcriptomic, and other clinical data.

In OS, the common subclassification method relies upon the characteristics of the dominant matrix present in the tumor. Conventionally, tumors occurring in fibrous tissue, bone, and cartilage are referred to as fibroblastic, osteoblastic, and chondroblastic OS, respectively [4,29]. These classes of tumors consist of about 75% of the cases; while the remaining 25% are considered to be variants, which show diverse biological properties. These variants are classified primarily based on the clinical factors, location of origin, and histological findings of the tumor. Similarly, the morphological and clinical feature-based classification includes parosteal, central low-grade, and periosteal OS. Tumors of these subclassifications constitute about 5% of the OS cases, but these have higher prognostic evaluations [30,31]. Despite an aggressive treatment regimen, the survival rate for OS is still a big challenge due to their propensity to metastasize before disease detection and to their resistance to therapies.

Unlike traditional histology-driven classifications, molecular subtyping of OS requires the identification of gene-based biomarkers and effective classifiers that can distinguish subtypes based on the molecular data. Prognostic biomarkers provide information about the patients’ overall cancer outcome and help to identify an appropriate therapeutic approach to cure or manage the disease. There is no systematic genome-wide analysis of these prognosis factors in OS [32,33,34,35]. Biomarker discovery in this area requires novel method development, which could improve the prognosis for OS patients. The current work focusses on developing novel gene-signature-based prognostic biomarkers in OS using multiomics data that include gene expression, DNA methylation, and miRNA expression data along with the associated clinical data. The findings may provide new insight in the domain of biomarker discoveries towards a better understanding of the subclassification of OS for developing improved treatment strategies.

## 2. Materials and Methods

**Multiomic data**: We downloaded gene expression, DNA methylation, miRNA expression, and clinical data for 85 common OS patients from Therapeutically Applicable Research To Generate Effective Treatments (TARGET, https://ocg.cancer.gov/, accessed on 1 December 2021) using a Bioconductor tool, TCGAbiolinks (an R/Bioconductor package for integrative analysis of TCGA data) [36]. We systematically accomplished data cleaning, global pattern analyses, and individual and integrative analyses on the multiomic datasets to remove noise and redundant data. Preprocessing steps of individual datasets are detailed in the following sections. Cox regression analysis and Kaplan Meier (KM) plots were generated to find genes associated with patient survival. Patient samples common across all data types were considered for downstream clustering analysis. Pathway analysis was carried out to identify the functional relevance of the differentially expressed genes between the identified molecular subtypes in OS.

**Protein coding and lncRNA data**: TCGAbiolinks was used to query and download the HTSeq–FPKM (Fragments Per Kilobase of Exon per Million) gene expression count data from the TARGET-OS project. Genes having more than 20% missing values across patients were not considered further for analysis. Additionally, only those genes with CPM (count per million) values > 1 for more than 10 % samples were retained. Genes were annotated as protein-coding or lncRNA using gencode reference annotation v22. The normalized count data were gene median centered, and the top 2500 protein-coding and 500 lncRNA based on median absolute deviation (MAD) were retained for further clustering analysis.

**miRNA data**: Normalized miRNA data were downloaded from NCI TARGET’s web portal. The dataset was first filtered by removing duplicated miRNA expression data. Further, clustering was performed using nonnegative matrix factorization (NMF) by retaining 190 (25%) of the most variable miRNAs.

**Methylation array data**: The raw IDAT files from the methylation study (NCBI GEO accession: GSE72872, Illumina HumanMethylation450 BeadChip array) [37] were downloaded using the R function, getGEOSuppFiles. The IDAT files were processed using the openSesame [38] tool in R to obtain the intensities of methylated and unmethylated alleles at the CpG sites as the beta value numeric. CpG probes with missing values in more than 20% samples across the patients were removed from further analysis. Imputation was performed to fill the missing β values in the data matrix using the imputeKNN function, with number of neighbor (k) of 15 [39]. This was followed by beta mixed-integer quantile (BMIQ) normalization using the ChAMP tool [40]. CpG probes mapped against sex and mitochondrial chromosomes were excluded from analyses to eliminate gender bias. CpG probes that overlapped with repeat masker and SNPs from dbSNP v151 with minor allele frequency (MAF) greater than 1% were removed to draw out sequence polymorphisms. For the final matrix, 5000 probes were retained based on the standard deviation for further downstream clustering analysis.

**Mutation data**: Mutation data from 123 OS patients was downloaded as a tab-delimited Mutation Annotation Format (MAF) file along with the scores, all-lesions, amplification, and deletion gene files generated by the Genepattern GISTIC module [35]. These data were used as input for the Bioconductor package, maftools [36]. Genes having a mutation in less than 5% of the samples were removed from further analysis. The matrix was converted into binary form for the presence or absence of mutations. The top 25% most variable genes (*n* = 1107) having a mutation in more than 20% of the samples were retained in the final matrix for clustering analysis.

**Clustering analysis**: NMF based clustering was performed using *nrun* = 500 for the number of clusters, *k* = 2 to 10 for miRNA, mutation, protein-coding, lncRNA, and methylation data for the same set of patients across all datasets by using R package NMF. The preferred clustering result was determined by using the observed cophenetic correlation coefficients between clusters and the average silhouette width of consensus cluster members.

**Cluster-Of-Clusters Analysis (COCA)**: The R package, COCA [41], an integrative clustering method was used to summarize clusters found in the protein-coding, lncRNA, miRNA, methylation, and mutation datasets by identifying a global clustering that is in good agreement with the clustering output in each of the individual datasets. The final clustering was accomplished based on the construction of the Matrix-Of-Clusters (MOC), a binary matrix of size N × K, where K is the sum of the number of clusters.

**Differential gene expression analysis**: Differential gene expression (DGE) analysis between the samples belonging to the two molecular subtypes obtained from the COCA was performed using the Bioconductor tool, DESeq2. The Benjamini-Hochberg (BH) adjusted *p*-value cut-off of 0.01, and an absolute log_2_Fold Change of 1 was used to obtain the list of differentially expressed genes. The heatmap was generated using tidyheatmap package in R.

**Pathway analysis**: All differentially expressed genes (with adjusted *p*-value < 0.01 and log_2_Fold Change = 1) between Subtypes 1 and 2 were used as input into QIAGEN IPA (QIAGEN, Redwood City, CA, USA; www.qiagen.com/ingenuity, accessed on 1 December 2021) for pathway enrichment analysis.

**Survival analysis**: We performed survival analyses for protein-coding, lncRNA, and miRNA genes using the log-rank test and Cox-regression analysis (*p*-value ≤ 0.05) by constructing high and low expression groups, using the median expression of genes as the cut-off value. For gene expression analysis, we used HTSeq–FPKM gene expression count data of OS patients. The R tool, “survival” was used for survival analysis, and the Kaplan–Meier survival curve plots were generated for all analyses. In addition, a log-rank or Mantel–Haenszel test was conducted to assess differences between survival curves of the two groups and to calculate the *p*-values. A hazard ratio (HR) > 1 indicates that patients in the high expression group had low survival, and HR < 1 suggested high or better survival.

**Correlation analysis**: Pearson correlation analysis between gene expression for the protein-coding genes was performed using the R cor.test function. The association was considered significant at a *p*-value < 0.05.

## 3. Results

In the present work, we have utilized the OS omics data of five distinct datatypes which include miRNA, mRNA, lncRNA, methylation, and mutation data, along with the survival data from public resources including the TARGET database and the PubMed repository. Initially, three data types (miRNA, mRNA, and lncRNA) along with the survival information were used to understand the associations of these biomolecules to the survival of OS patients. The later part of the study mainly focused on COCA-based clustering of 85 patients using multi-level omics information including miRNA, mRNA, lncRNA, methylation, and mutation, resulting in the identification of two OS subtypes. Results from each analysis are described in the following sections.

### 3.1. Mucin Family Genes Are Highly Mutated in OS

For mutational signature analysis, we read MAF files using maftool, which provided frequencies of six classes of the mutation types (C > A, C > G, C > T, T > A, T > C, and T > G) in each sample. The contribution of each mutational signature was analyzed across the samples. Our analyses showed that the commonly mutated genes include *PABPC3*, *HRNR*, *KMT2C*, and *PRSS3* in more than 65% of the samples (Appendix A). A missense mutation was found to be the most common variant type across all patients, although many of the genes were found to have multiple mutations as Multi_Hit (green), as indicated in Appendix A. The analysis also showed that the top-ranked altered genes include the MUC family of genes, namely, *MUC6* (80%), *MUC12* (76%), and *MUC4* (76%). An oncoplot of members of the MUC family genes with their mutation frequencies is shown in Appendix A.

Enriched pathways associated with OS genetic alterations were analysed using the OncogenicPathways module in maftools for known signaling pathways in TCGA cancers [42]. These include the P53, Notch, Myc, Wnt, Hippo, RTK-RAS, PI3K, and Cell cycle signaling pathways. The top five most commonly altered genes prevalent in more than 10% of the samples containing mutations in the TCGA enriched pathways are shown in Figure 1A. The genes associated with these pathways include *TEAD2*, *WWC1*, *AJUBA*, *CSNK1D*, and *LLGL2* in hippo signaling, *IGF1R*, *RAC1*, *ERBB4*, *GRB2*, and *NF1* in RTK-RAS pathway, RFNG, LFNG, MAML3, APH1A, and NOTCH2 in Notch pathway, and *WNT7B*, *LZTR1*, *DKK1*, *APC*, and *FZD6* in Wnt pathway. In the TP53 pathway, genes including *CHEK2*, and *ATM* mostly showed missense mutations (blue color), while the *TP53* gene had multiple mutations including missense, Amplification (Amp), Deletion (Del), and Frameshift. NOTCH signaling pathways mostly showed Amp mutation as presented in brick color. Similarly, the genes *LLGL2*, *CSNK1D*, and *AJUBA* in the HIPPO pathway showed Amp mutation; while other genes, *WWC1* and *TEAD2* in the same pathway, were deleted. Cell cycle pathways include genes RB1 which showed multiple mutation types, whereas, other genes *CCNE1* and *CDK4* showed mainly amplification. Multiple types of mutation were identified in *RB1* in Cell Cycle and *TP53* in TP53 pathways, *NAML3* in NOTCH pathways, *MGA* in MYC pathways, and *NF1* in RTK-RAS pathways. In general, amplifications were most prevalent across pathways. The fraction of affected genes in the most enriched pathways included 94% in Hippo, 86% in Wnt, 84% in RTK-RAS, and 84% in Notch signaling pathways (Figure 1B). The pathways, associated genes, fraction of mutated samples, and other related information is provided in Appendix A.

### 3.2. Survival Analysis Reveals Key Prognostic Pathway Genes

Survival analysis of OS patients was performed using gene expression, miRNA, and lncRNA, with *p*-values < 0.05 for both log-odd and Cox regression. In-house R code with survival and survminer packages in the background were used to perform this analysis.

The log-Rank test and cox-regression analysis (*p*-value < 0.05) using protein-coding HTSeq-FPKM gene expression identified 430 genes associated with OS prognosis (*p*-value < 0.01). The genes with HR more than 4.5 include *RAMP1*, *TAC4*, *MT1A*, *CRIP1*, *NHEJ1*, *TIMM23B*, *COL5A2*, *UFC1*, and *CORT*. The higher expression of these genes indicate low survival of OS patients. In contrast, higher expression of genes such as *METTL20*, *MYH10*, *PDE1B*, *CCDC42*, and many more which got HR < 1 refers to better survival rates. In each category, survival plots for the top three potential genes are provided in Figure 2A,B, respectively. The HR and *p*-value of all 430 genes are provided in Appendix A subset of the top 49 genes, significantly (*p*-value < 0.001) related to the survival of OS patients, are shown in Table 1.

We mapped the expression of 34 genes involved in 10 canonical pathways that are considered to be the likely cancer drivers (functional contributors) or therapeutic targets [42], and are found to be associated with the survival of patients (*p* < 0.05) in our analysis. As shown in Figure 3A, most of the genes had either positive or no correlation within the pathways at *p* < 0.05. Only a few genes such as *SOST* and *WNT5A* in WNT signaling pathways, *MAPK1* and *PEBP1* in RTK-RAS pathways, and *PIK3R3* and *RPS6* in PI3K pathways, showed negative expression pattern within the pathways. In our analyses, none of the genes in the NRF2 and TGFβ signaling pathways were significant in survival and are, therefore, not shown in Figure 3A. HR values of 34 genes from these pathways are shown in Figure 3B. Genes including *ATM* in TP53 pathway, *CDKN2A* in Cell cycle, *FAT1* and *FAT3* in HIPPO signaling, *MXl1* in MYC pathway, *DTX3*, *DTX2*, *JAG2*, *HES4*, and *HES5* in NOTCH signaling showed HR values greater than 1 indicating their association with low survival in OS patients. Other genes such as *LRP5*, *SOST*, *WNT1*, *RPS6* in different pathways also have HR value more than 1 with possible association with survival (Figure 3B). In contrast, genes such as *KRAS*, *WNT5A*, and *CDK6* have a HR less than 1 indicating their association with better survival in OS patients.

Survival analysis in relation to lncRNAs showed that 36 lncRNAs (*p*-value < 0.01) or 142 lncRNAs (*p*-value < 0.05) are correlated with survival. ELFN1-AS1 (HR > 1) is among the top 3 lncRNAs identified (*p*-value < 0.001) in our study, which has been recognized as an important signature to predicting OS patient survival in a previous study [43]. Similarly, RP11-283C24.1 and lnc01060 also displayed HR > 1, indicating that higher expression of these lncRNAs may lead to lower survival of OS patients. No information was found in the literature about two other lncRNAs, lnc01060 and RP11-283C24.1 and further investigations are required to validate these potential biomarkers. Our analysis also identified lncRNAs CTD-2078B5.2, RP11-78O7.2, and RP11-446N19.1, which got HR < 1 and *p*-value < 0.05. Higher expression of these lncRNAs are associated with better survival of the patients; hence, they serve as potential biomarkers for OS prognosis. Survival plots of the top lncRNAs with HR > 1 and HR < 1 are provided in the Appendix A. The HR and beta values of all 36 and 142 lncRNAs identified above are provided in the Appendix A.

In addition, survival analysis using miRNA data resulted in the identification of 76 (*p* < 0.05) or 31 miRNAs (*p* < 0.01), which showed association with the survival (Appendix A). We identified four miRNAs, miR-122, miR-200b, miR-1298, and miR-1264 with HR < 1 and *p*-value < 0.05. Similarly, miRNAs with HR > 1 and *p*-value < 0.05 include hsa-miR-182, hsa-miR-891a, hsa-miR-190, miR-452, hsa-miR-142, and 62 more. Similar to mRNAs and lncRNAs, higher expression of miRNAs with HR > 1 indicate lower survival of the patients; whereas, higher expression of miRNAs with HR < 1 shows better survival association with the OS patients. Previous studies have shown that miRNA, hsa-miR-190 was involved in many cancer-related biological processes such as metastasis, proliferation, and apoptosis by means of dysregulating target genes [44,45,46]. Similarly, hsa-miR-452 and hsa-miR-122 were found to be involved in promoting colorectal and colon cancer progression, respectively [47,48]. The Kaplan-Meier survival plots for top miRNAs are shown in Appendix A.

### 3.3. Clustering Analysis

Clustering analysis with NMF was performed for 85 patients using the miRNA, mutation, protein-coding, lncRNA, and CpG probe methylation datasets. The final matrix consisted of 190 miRNAs, 1107 mutations, 2500 protein-coding regions, 500 lncRNA, and 5000 CpG methylation sites, which were obtained after the preprocessing steps as described previously in the methods section. The optimal number of clusters was obtained by evaluating the cophenetic correlation coefficients between clusters and the average silhouette width (ASW), resulting in three clusters each for protein-coding, lncRNA, miRNA, and methylation data, and four clusters for the mutation data (Figure 4).

A matrix of clusters was generated and used as input for COCA, which resulted in two major integrated molecular clusters/subtypes of OS, consisting of 44 and 41 patients in Subtype-1 and Subtype-2, respectively. The matrix of clusters was plotted using plotMOC function. Appendix A.

### 3.4. Comparison of the Identified Molecular Subtypes

Differential Gene Expression analysis was performed between the two molecular subtypes obtained from COCA using the Bioconductor package, DESeq2 [49]. A total of 261 genes were differentially expressed between Subtype-2 vs. Subtype-1 at log_2_Fold Change > 1.5 and adjusted *p*-value < 0.05 (Figure 5A). Among these, 178 were upregulated and 83 were downregulated. A heatmap of the top 100 most variables differentially expressed genes is shown in Figure 5B. Hierarchical clustering based on gene expression data showed two distinguishable molecular subtypes consistent with the multiomics-based subclusters. The expression-based heatmap (Figure 5B) clearly showed the differences in gene expression levels between Subtype 1 (blue color) and Subtype 2 (red color). We utilized gene count data from DEG’s for clustering using euclidian distance measure for hierarchical clustering with tidyheatmap library in R, which also resulted in two major clusters consistent with the multiomics-based molecular subtyping (Appendix A, Figure 5B). However, no genes were found to be significantly differentially mutated between the two molecular subtypes. The pathway enrichment analysis of survival associated protein-coding genes (as identified in log-odd test and Cox-regression analysis, *p*-value < 0.05) using IPA identified WNT signaling and Axonal Guidance signaling pathways, which are related to the OS pathogenesis [50,51,52] (Appendix A). However, no significant difference was observed in the survival of patients between the two subtypes.

## 4. Discussion

OS is one of the most frequent and aggressive malignant bone neoplasms. Identification of clinically relevant prognostic markers in OS is of vital importance. Cancer subtyping studies using multiomics data have been widely performed to get an understanding of the genetic mechanism of the disease as well as to accelerate their therapeutic applications [53,54,55,56]. Although, many markers have been demonstrated to be of prognostic significance in the treatment of OS, there is a need for new therapeutic targets that can be evaluated [33,57]. In the current study, we accomplished global analysis of different types of omics data from the OS patients in the TARGET database, and identified molecular subtypes by integrative analysis of multiomics data.

Mutation analysis identified significant mutations in the Mucin (MUC) family of genes, notably *MUC6* (80% of the patients), *MUC12* (76% of the patients), and *MUC4* (76% of the patients). Previously, Tirabosco et al. have reported MUC4 as the immunoreactive gene in fibromyxoid and sclerosing tumors in bone [58]. Similarly, *MUC12* and *MUC4* were identified as top-altered genes in a cohort of 21 OS patients [59]. However, the understanding of whether or not these mutations in MUC genes are driving OS is still unknown. In our analyses, *MUC15* which promote cell proliferation, migration, and invasion [60], mutated in only 2% of the TARGET OS patient cohort.

Further, the gene *KMT2C*, which is mutated in 67% of all OS samples, involved in OS carcinogenesis and progression [61]. Similar to results of our study, the exome analysis shows a high load of somatic variations in *KMT2C*, which can have histone modification-related activity in OS [62]. Further investigation about the roles of identified mutations in various genes could facilitate the development of prognostic biomarkers for OS. Subsequent pathway-level analysis showed that the Wnt pathway is one of the top enriched pathways in OS samples and is significantly involved in OS progression, as supported by several previous studies [50,63,64].

We also observed enrichment of pathway members that contained recurrent or known alterations that are likely to be cancer drivers or therapeutic targets in the Hippo, RTK-RAS, Notch, and Wnt signaling pathways [42] (Figure 1). The Hippo and Notch signaling pathways have an active role in OS [65,66,67]. Fang et al., 2018 have shown that the hyperactive Wnt/β-catenin pathway may be required for proliferation and metastasis of OS [68]. In addition, similar to the present results, somatic mutations and upregulated gene expression of many components in the Wnt/β-catenin pathway were observed in their study.

Survival analysis revealed the survival association of several genes to OS including those with high expression such as the receptor activity-modifying proteins 1 (*RAMP1*): a protein of calcium signaling receptor complex; Tachykinin Precursor 4 (*TAC4*), Metallothionein 1A (*MT1A*), *CRIP1*, and *CORT*. Our analysis showed that these genes are associated with low survival of OS patients based on their HR values (*RAMP1*: 5.7, *TAC4*: 5.5, *MT1A*: 4.8, *CRIP*: 4.8, *CORT*: 4.6) (Figure 2B, Appendix A). Similarly, *CHST13* (HR: 2.4) is associated with low survival, while *DDX60L* (HR 0.37), *METTL20* (HR 0.14), *MYH10* (HR 0.16), and *PDE1B* (HR 0.15) are associated with higher survival outcomes in OS patients (Figure 2A, Appendix A). These genes have the potential to serve as prognostic biomarkers in OS. The prognostic role of *RAMP1* in OS has not been reported in the literature. *RAMP1* interacts with G-protein-coupled receptors, such as calcitonin receptor-like receptor, calcitonin receptor, calcium-sensing receptor, and glucagon receptor [69,70]. Knockdown of *RAMP1* reduced clonogenic/spheroidal growth and tumorigenicity, and small-molecule inhibitors directed against the *RAMP1* reduced growth of Ewing sarcoma [69].

A lower survival rate in patients was related to high *TAC4* and *MT1A* expression. Although the role of *TAC4* in metastasis and prognosis of OS is unknown, other bioinformatic studies have shown a possible link [71]. *MT1A* expression levels were used as one of the genes to develop a machine learning-based prognostic risk model in OS patients, suggesting that this gene is associated with OS survival [72].

From our study, higher expression of *CRIP1* and *CORT* was associated with lower survival (*p* < 0.001). *CRIP1* has been previously shown to be over-expressed in pre-therapeutic OS samples [73]. *CORT* is an endogenous cyclic neuropeptide known to regulate the growth and metastasis of lung and thyroid cancer [74,75]. The higher expression of *CORT* gene has also been associated with high-risk group OS patients [76]. In addition, we have also identified lncRNA and miRNA which were associated with OS survival. In particular, higher expression of miRNAs hsa-miR-190, hsa-miR-452, and hsa-miR-142 are directly associated with lower survival of the OS patients. In contrast, hsa-miR-122, hsa-miR-200, and hsa-miR-1298 had association with improved survival of the patients. Similarly, lncRNAs ELFN1-AS1, lnc01060, RP11-283C24.1, and RP11-283C24.1 with HR > 1 and lncRNAs CTD-2078B5.2, RP11-78O7.2, and RP11-446N19.1 with HR < 1 were interpreted to be associated with lower and higher survival of the OS patients. The experimental validation is required to support these findings.

Our global clustering strategy built upon the multiomics landscape has identified two main subtypes (Subtype 1: 44 patients, Subtype 2: 41 patients), which opens up the potential for developing effective treatments. Differential expression and pathway enrichment analyses suggest that both subtypes had different deregulated pathways associated with OS pathology. We observed over 250 differentially expressed genes between two subtypes (log_2_Fold Change > 1.5 and adjusted *p*-value < 0.05), including genes associated with Wnt signaling, calcium signaling, and PI3K-Akt signaling pathways, were highly enriched and found related to OS pathogenesis.

Although no difference in somatic mutation and somatic copy number alteration levels between the identified subtypes, hierarchical clustering of the top 100 most differentially expressed genes between the two OS subtypes could be used to classify the patients into two molecular clusters. Liu et al. 2020 [1] identified differentially regulated genes i.e., *VAMP8*, *A2M*, *HLA-DRA*, *SPARCL1*, *HLA*-*DQA1*, *APOC1*, and *AQP1*, which are involved in metastasis of OS. Our analysis showed that genes *VAMP8*, *SPARCL1*, and *APOC1* were significant with *p*-value 0.007, 0.017, and 0.038, respectively, for predicting survival in OS patients. Our analysis also identified three upregulated genes, *POP4*, *HEY1*, *CERKL*, and seven other downregulated genes, *CEACAM1*, *ABLIM1*, *LTBP2*, *ISLR*, *LRRC32*, *PTPRF*, and *GPX3* between identified two subtypes.

## 5. Conclusions

This study has identified potential prognostic genes including *CRIP1*, *CORT*, *CHST13*, and *DDX60L* in the OS samples. Although many other cancer-related genes showed a high frequency of mutations in OS, the MUC family of genes (*MUC6*, *MUC12*, and *MUC4*) were found to be highly mutated in our study. miRNAs: hsa-miR-190, hsa-miR-452, hsa-miR-142, hsa-miR-122, and hsa-miR-200, and lncRNAs: ELFN1-AS1, lnc01060, RP11-283C24.1, RP11-283C24.1, CTD-2078B5.2, RP11-78O7.2, and RP11-446N19.1 were also found to be significantly associated with the survival of OS patients; and therefore, interpreted as potential biomarkers that can to be further validated with experimental studies. Our detailed clustering approach using COCA has identified two main OS subtypes, which are significantly different at the molecular level. These findings offer a strong basis to understand the genetic heterogeneity of OS and develop next generation prognostic biomarkers for effective treatment of OS patients.

## Figures and Tables

**Figure 1 cancers-15-02134-f001:**
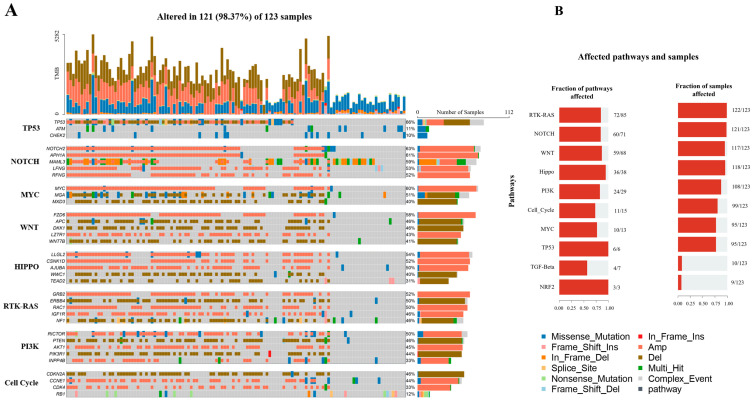
(**A**) Oncoplot showing the top 5 altered genes with more than 10% samples containing mutations in the TCGA enriched pathways. The fraction and the number of samples with variations in the corresponding genes are shown on the right as percentages and the bar plots (**B**) Enriched oncogenic pathways showing the fraction of genes and samples affected.

**Figure 2 cancers-15-02134-f002:**
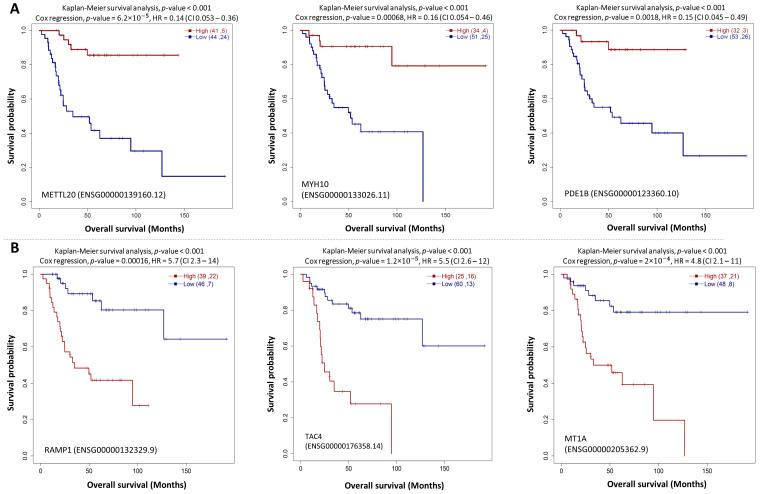
Kaplan-Meier plot for high vs low expression group in TARGET OS data with *p*-value from log-rank test and Cox regression model for genes, *METTL20*, *MYH10*, and *PDE1B* with HR < 1 (**A**), and genes, *RAMP1*, *TAC4*, and *MT1* with HR > 1 (**B**).

**Figure 3 cancers-15-02134-f003:**
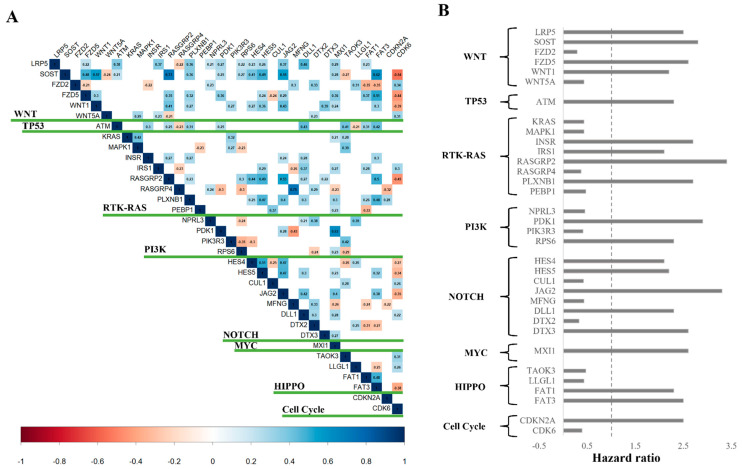
(**A**) Correlation plot showing significantly correlated canonical pathway genes (*p* < 0.05). Negative correlated genes are shown in red, and positive in blue. (**B**) List of 34 genes found significantly associated with survival (*p* < 0.05), their associated canonical pathways, and corresponding HR values.

**Figure 4 cancers-15-02134-f004:**
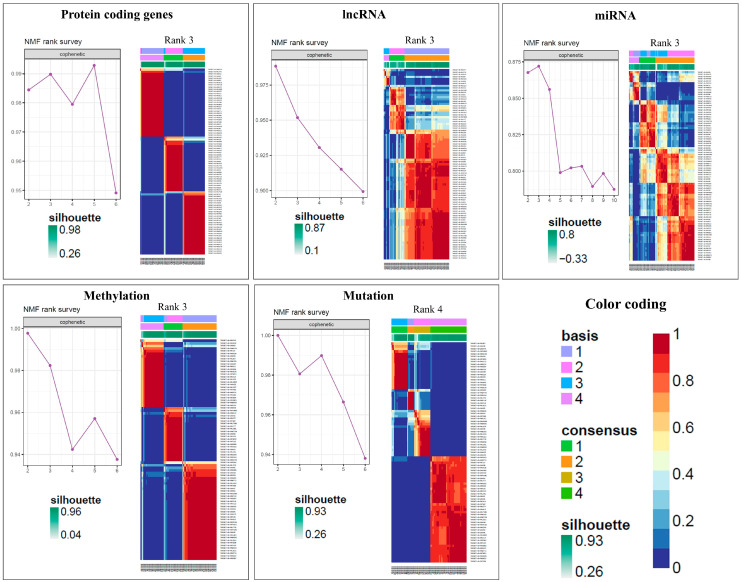
Average silhouette width and NMF classification with rank using 190 miRNAs, 1107 mutations, 2500 protein-coding regions, 500 lncRNA, and 5000 CpG methylation probe data for 85 OS-TARGET patients which was further used as input for COCA for obtaining the final two molecular subtypes.

**Figure 5 cancers-15-02134-f005:**
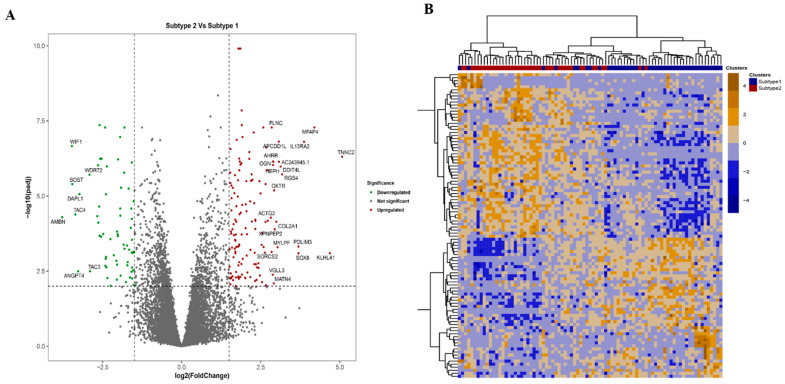
(**A**) Volcano plot showing the differentially expressed genes between the two molecular subtypes. The downregulated (green) and upregulated (red) genes obtained at abs(log_2_Fold Change) > 1.5 and adjusted *p*-value < 0.05 are highlighted. (**B**) Heatmap showing top 100 most variable differentially expressed genes obtained at abs(log_2_Fold Change) > 1 and adjusted *p*-value < 0.01 cutoff found between the two molecular clusters.

**Table 1 cancers-15-02134-t001:** List of the top 49 significant (*p*-value < 0.001) genes found in survival analysis. The gene list is sorted based on beta values. HR: Hazard Ratio.

Gene	HR	HR_low95	HR_up95	Beta	Gene	HR	HR_low95	HR_up95	Beta
*RAMP1*	5.7	2.3	14	1.7	*LY86*	0.24	0.1	0.56	−1.4
*TAC4*	5.5	2.6	12	1.7	*SHISA5*	0.24	0.1	0.57	−1.4
*TIMM23B*	4.7	2.1	11	1.6	*FOLR2*	0.25	0.11	0.56	−1.4
*COL5A2*	4.7	2	11	1.6	*C11orf45*	0.25	0.11	0.57	−1.4
*MT1A*	4.8	2.1	11	1.6	*UBE2L3*	0.24	0.11	0.54	−1.4
*CRIP1*	4.8	1.9	12	1.6	*CYFIP1*	0.24	0.1	0.55	−1.4
*UFC1*	4.6	1.9	11	1.5	*SNX1*	0.22	0.091	0.55	−1.5
*PROSER2*	4.4	1.9	10	1.5	*ACTB*	0.22	0.095	0.5	−1.5
*NHEJ1*	4.7	2	11	1.5	*MPP1*	0.22	0.091	0.55	−1.5
*CORT*	4.6	2.1	10	1.5	*CD180*	0.23	0.092	0.56	−1.5
*CKMT2*	4.1	2	8.8	1.4	*TPMT*	0.22	0.09	0.55	−1.5
*CHD1L*	3.9	1.7	8.9	1.4	*BBS4*	0.22	0.088	0.53	−1.5
*LGR6*	4.1	1.9	9.1	1.4	*ITGAM*	0.22	0.091	0.56	−1.5
*IFFO2*	4.2	1.8	9.7	1.4	*TCN2*	0.23	0.093	0.56	−1.5
*MAFK*	4.1	1.8	9.7	1.4	*SIRPA*	0.21	0.087	0.53	−1.5
*PGAM4*	3.9	1.8	8.5	1.4	*SETD9*	0.19	0.079	0.48	−1.6
*SNAP91*	3.8	1.8	8	1.3	*SNTB2*	0.21	0.085	0.53	−1.6
*PDE4C*	3.5	1.7	7.5	1.3	*MITF*	0.21	0.079	0.55	−1.6
*KRT2*	3.6	1.7	7.6	1.3	*NUBP1*	0.17	0.073	0.41	−1.8
*GCSAM*	3.7	1.7	8	1.3	*IRF2BPL*	0.17	0.069	0.42	−1.8
*BAI1*	3.7	1.7	7.8	1.3	*ERCC4*	0.17	0.066	0.46	−1.8
*CFAP44*	3.8	1.8	8	1.3	*PDE1B*	0.15	0.045	0.49	−1.9
*FAM166B*	3.5	1.7	7.3	1.3	*MYH10*	0.16	0.054	0.46	−1.9
*APBB1IP*	0.24	0.11	0.55	−1.4	*METTL20*	0.14	0.053	0.36	−2
*COMMD9*	0.24	0.11	0.55	−1.4					

## Data Availability

All analyses were performed using the R version 4.0.1 (R Development Core Team 2015). We performed clustering, pathway and survival analysis by using R/-Bioconductor tools. All the relevant data generated in the project is made available in the Appendix A.

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
