# Peer review of "Molecular Subtyping and Survival Analysis of Osteosarcoma Reveals Prognostic Biomarkers and Key Canonical Pathways"

_cancers, 2023, doi:10.3390/cancers15072134_

Round 1

Reviewer 1 Report

Comments on cancers-2294731

 In this study, the author has studied “Molecular subtyping and survival analysis of osteosarcoma reveals prognostic biomarkers and key canonical pathways.This is an engaging article with a robust approach that purposefully questions our knowledge of the subject. However, the presentation of the methodology is somewhat confusing, and the readability of the discussion could be improved. Addressing both these issues will make this interesting paper more impactful. The English language used in the manuscript needs some improvements as some punctuation, and grammatical mistakes are present. Experimental designs required more clarity.

Specific comments:

1.      The authors are advised to revise the abstract, especially the results section. 

2.      Please add more strong keywords and avoid the words used in the title.

3.      Page 4, line 183-184: “In the present work, we have utilized the OS omics data of five distinct datatypes, i.e.. miRNA, mRNA…” It is suggested to avoid ‘e.g., i.e., etc.,’ in the scientific manuscript.

4.      The discussion also needs to be revised. The discussion needs professional English editing, and please revise them carefully to make it standard. Please focus on the main topic during the discussion. An excellent discussion contains an accurate statement of the results, the relevance, and importance of the results, suitable comparisons to similar studies, alternative explanations of the findings, known limitations, and suggestions for future research.

5.      It is suggested to add a separate heading for concussion.

6.      Authors are advised to proofread the manuscript to overcome grammatical mistakes.

7.      Authors are advised to revise a few subheadings.

8.      It is suggested to add the latest data in the manuscript. Only 3-4 references from 2022 are present in the manuscript. Many manuscripts containing the same research theme are recently published.

Author Response

Response to Reviewer Comments:

We thank all the reviewers for their valuable suggestions to improve the quality of the work presented in this manuscript. We have revised the manuscript to address all the reviewers’ concerns New changes were highlighted in yellow throughout the manuscript. Below, we provide a point-by-point response to reviewers’ comments/suggestions.

Reviewer-1

In this study, the author has studied “Molecular subtyping and survival analysis of osteosarcoma reveals prognostic biomarkers and key canonical pathways.” This is an engaging article with a robust approach that purposefully questions our knowledge of the subject. However, the presentation of the methodology is somewhat confusing, and the readability of the discussion could be improved. Addressing both these issues will make this interesting paper more impactful. The English language used in the manuscript needs some improvements as some punctuation, and grammatical mistakes are present. Experimental designs required more clarity.

Specific comments:

1. The authors are advised to revise the abstract, especially the results section.

Response: As suggested, the abstract was revised by covering all the major results.

2. Please add more strong keywords and avoid the words used in the title.

Response: We replaced some keywords with stronger ones. (Lines 42-43)

3. Page 4, line 183-184: “In the present work, we have utilized the OS omics data of five distinct datatypes, i.e.. miRNA, mRNA…” It is suggested to avoid ‘e.g., i.e., etc.,’ in the scientific manuscript.

Response: Sentence was rewritten by removing abbreviations such as e.g., i.e. etc. We also corrected the text throughout the manuscript by avoiding the use of such instances.

4.      The discussion also needs to be revised. The discussion needs professional English editing, and please revise them carefully to make it standard. Please focus on the main topic during the discussion. An excellent discussion contains an accurate statement of the results, the relevance, and importance of the results, suitable comparisons to similar studies, alternative explanations of the findings, known limitations, and suggestions for future research.

Response: As suggested, we revised the discussion and conclusion sections and also edited the entire manuscript with some help from a native English speaker. In the discussion, we added new text corresponding to the identified gene, miRNA and lncRNA biomarkers, as suggested by Reveiwer-2. (Major changes: Lines 386-387, 407-411; Minor changes: as highlighted)

5. It is suggested to add a separate heading for conclusion.

Response: Separate ‘Conclusion’ section was added as suggested. (Lines 431-443)

6. Authors are advised to proofread the manuscript to overcome grammatical mistakes.

Response: The manuscript was proofread for grammatical errors and sentences were corrected throughout the manuscript.

7. Authors are advised to revise a few subheadings.

Response: As suggested, we have revised some subheadings mainly in the methods section.

8. It is suggested to add the latest data in the manuscript. Only 3-4 references from 2022 are present in the manuscript. Many manuscripts containing the same research theme are recently published.

Response: Thanks for pointing this out. We have revised the text and included the following latest references. Rédini et al.(2022), Liu et al.(2021), Wang et al.(2022), Zhao et al.(2022), Ochoa et al.(2022), Qi et al.(2021), Ben-Ghedalia-Peled et al.(2022), and Zhang et al.(2023)

Reviewer 2 Report

The authors used numerous bioinformatics tools to combine publicly available multinomics data such as ncRNA and mRNA expression, DNA methylation and clinical outcomes in osteosarcoma to identify two molecularly distinct tumor subtypes, and these results might help find new diagnostic and therapeutic biomarkers and novel treatment regimens for osteosarcoma.

The introduction part is written well and contains sufficient information regarding the field of study. Materials and methods are well described, methodology is correct. Several minor issues in other paragraphs of the manuscript needs to be addressed:

Paragraph 3.2. contains Kaplan-Meier survival plots for three genes involved in overall survival, with the highest HR, meaning that high expression of these genes drastically shortens patients survival and can be used as prognostic biomarkers in osteosarcoma. What about the genes with HR < 1, such as METTL20, MYH10, PDE1B or other osteosarcoma-related genes? Can their high expression be also an indicator of higher possibility of patients survival? Can the authors add Kaplan-Meier plots also for these three genes with the lowest HR index?

Further the authors suggest that lncRNA ELFN1-AS1 as well as four miRNAs: miR-190, miR-452, miR-122 and miR-1298 might also be considered as prognostic biomarkers in osteosarcoma. Kaplan-Meier plots for these ncRNAs should be also added to the manuscript, and the involvement of these ncRNAs in osteosarcoma should be thoroughly deliberated in the discussion section.

To be honest, I don’t really understand the main conclusion or the notion of the part when two osteosarcoma subtypes have been identified. What is the main bottom line for this discovery? How are these subtypes different and what is the clinical significance of them? How can these subtypes be easily identified in clinical environment? Are these subtypes different in treatment response or patient survival? Are the identified here potential biomarkers of osteosarcoma behaving in the same manner in both subtypes or are there any differences? The authors should rewrite some parts of the discussion section and highlight the impact of the discovery.

Author Response

Response to Reviewer Comments:

We thank all the reviewers for their valuable suggestions to improve the quality of the work presented in this manuscript. We have revised the manuscript to address all the reviewers’ concerns New changes were highlighted in yellow throughout the manuscript. Below, we provide a point-by-point response to reviewers’ comments/suggestions.

Reviewer-2

1. The authors used numerous bioinformatics tools to combine publicly available multinomics data such as ncRNA and mRNA expression, DNA methylation and clinical outcomes in osteosarcoma to identify two molecularly distinct tumor subtypes, and these results might help find new diagnostic and therapeutic biomarkers and novel treatment regimens for osteosarcoma.

Response: Thank you for the feedback.

2. The introduction part is written well and contains sufficient information regarding the field of study. Materials and methods are well described, methodology is correct. Several minor issues in other paragraphs of the manuscript needs to be addressed:

Response: Thank you for pointing this out. We have revised the text thoroughly throughout the manuscript and edits from a native English speaker were included. Please see the tracked and highlighted changes throughout the manuscript.

3. Paragraph 3.2. contains Kaplan-Meier survival plots for three genes involved in overall survival, with the highest HR, meaning that high expression of these genes drastically shortens patients survival and can be used as prognostic biomarkers in osteosarcoma. What about the genes with HR < 1, such as METTL20, MYH10, PDE1B or other osteosarcoma-related genes? Can their high expression be also an indicator of higher possibility of patients survival? Can the authors add Kaplan-Meier plots also for these three genes with the lowest HR index?

Response: Thanks for the advice. As suggested, we included the survival plots for genes, METTL20, MYH10, and PDE1B in the main text. Figure 2 is changed to 2A (genes with HR < 1) and 2B (genes with HR >1). We also added the relevant text in the results and discussion sections corresponding to these changes. (Figure 2A & B, Lines 244-248, 385, 387-388)

4. Further the authors suggest that lncRNA ELFN1-AS1 as well as four miRNAs: miR-190, miR-452, miR-122 and miR-1298 might also be considered as prognostic biomarkers in osteosarcoma. Kaplan-Meier plots for these ncRNAs should be also added to the manuscript, and the involvement of these ncRNAs in osteosarcoma should be thoroughly deliberated in the discussion section.

Response: We have included survival plots for miRNAs: miR-122, miR-200b, miR-1298, and miR-1264 with HR < 1 and p-value < 0.05. Similarly, survival plot for miRNAs with HR > 1 and p-value < 0.05 were also added. We added a new Supplementary Figure S4. Relevant text was added in the Results (Lines 293-298), Discussion (Lines 405-408), and Conclusion (Lines 435-436) sections.

Similarly, survival plots for lncRNAs, ELFN1-AS1, RP11-283C24.1, and lnc01060 (HR > 1) and lncRNAs: CTD-2078B5.2, RP11-78O7.2, and RP11-446N19.1 (HR <1) were added in Supplementary Figure S3. We included the text corresponding to these changes in the Results (Lines 281-289), Discussion (Lines 409-411), and Conclusion (Lines 436-437) sections.

5. To be honest, I don’t really understand the main conclusion or the notion of the part when two osteosarcoma subtypes have been identified. What is the main bottom line for this discovery? How are these subtypes different and what is the clinical significance of them? How can these subtypes be easily identified in clinical environment? Are these subtypes different in treatment response or patient survival? Are the identified here potential biomarkers of osteosarcoma behaving in the same manner in both subtypes or are there any differences? The authors should rewrite some parts of the discussion section and highlight the impact of the discovery.

Response: We have characterized identified sub-groups based on differential gene expression analysis. No significant differences in the survival or mutation profiles were observed between the two groups. However, pathway enrichment showed differences in the genes/miRNA/lncRNA which are associated with survival of OS patients. (Paragraph, Lines 421-430)

Round 2

Reviewer 1 Report

The authors have carefully revised the comments. So, the manuscript should be accepted in the present form.